# Oligogalacturonides Enhance Resistance against Aphids through Pattern-Triggered Immunity and Activation of Salicylic Acid Signaling

**DOI:** 10.3390/ijms23179753

**Published:** 2022-08-28

**Authors:** Christian Silva-Sanzana, Diego Zavala, Felipe Moraga, Ariel Herrera-Vásquez, Francisca Blanco-Herrera

**Affiliations:** 1Centro de Biotecnología Vegetal, Facultad de Ciencias de la Vida, Universidad Andres Bello, Santiago 8370146, Chile; 2Millennium Science Initiative Program (ANID), Millennium Nucleus for the Development of Super Adaptable Plants (MN-SAP), Santiago 8370186, Chile; 3Millennium Science Initiative Program (ANID), Millennium Institute for Integrative Biology (iBio), Santiago 8370186, Chile; 4Center of Applied Ecology and Sustainability (CAPES), Santiago 8320000, Chile

**Keywords:** oligogalacturonides, aphid, plant defense, pattern-triggered immunity, reactive oxygen species, callose, salicylic acid

## Abstract

The remarkable capacity of the generalist aphid *Myzus persicae* to resist most classes of pesticides, along with the environmental and human health risks associated with these agrochemicals, has necessitated the development of safer and greener solutions to control this agricultural pest. Oligogalacturonides (OGs) are pectin-derived molecules that can be isolated from fruit industry waste. OGs have been shown to efficiently stimulate plant defenses against pathogens such as *Pseudomonas syringae* and *Botrytis cinerea*. However, whether OGs confer resistance against phytophagous insects such as aphids remains unknown. Here, we treated *Arabidopsis* plants with OGs and recorded their effects on the feeding performance and population of *M. persicae* aphids. We also identified the defense mechanism triggered by OGs in plants through the analysis of gene expression and histological approaches. We found that OG treatments increased their resistance to *M. persicae* infestation by reducing the offspring number and feeding performance. Furthermore, this enhanced resistance was related to a substantial accumulation of callose and reactive oxygen species and activation of the salicylic acid signaling pathway.

## 1. Introduction

Aphids are a primary agricultural concern because they transmit viral diseases and reduce crop yield by feeding on phloem sap [1,2,3]. Agrochemicals are commonly used to control aphids. However, these insects have evolved resistance to most classes of insecticides [4,5] and have become one of the costliest pests in terms of pesticide applications [6]. In addition, the excessive use of insecticides adversely affects human health and pollutes the environment [7,8]. Thus, developing greener alternatives to control crop pests is necessitated.

Oligogalacturonides (OGs) are pectin-derived oligomers that can be extracted from fruit industry waste, such as citrus peel and apple pomace [9,10,11]. They have been shown to enhance the resistance of plants against relevant pathogens, such as *Botrytis cinerea*, *Pectobacterium carotovorum*, and *Pseudomonas syringae* [12,13,14,15]. OGs are naturally produced by plants following infection by pathogens harboring pectin-modifying enzymes, such as pectin methylesterases, polygalacturonases, and pectate lyases, which break down the homogalacturonan chains that constitute pectin to release OGs. The extracellular pectin-binding domain of the wall-associated kinases (WAKs) recognizes OGs, triggering a defense response through the mitogen-activated protein kinase signaling cascade [16,17,18,19], which involves common pattern-triggered immunity (PTI) responses, such as accumulation of reactive oxygen species (ROS) and deposition of callose [12,13,20,21,22]. OGs also increase the transcription of defense genes related to the salicylic acid (SA) and jasmonic acid signaling pathways [14,15]. Moreover, the defense responses triggered by OGs can vary depending on their degree of polymerization (DP), i.e., the number of galacturonic acid residues that constitute the oligomer. For instance, in *Arabidopsis*, longer OGs (DP = 10–15) induce higher levels of ROS synthesis and defense-related gene transcription than shorter OGs (DP = 3) [20].

Although OGs increase plant resistance to pathogens, whether they confer the same benefits against aphids remains unknown. Therefore, this study aimed to investigate the potential protective effect of OGs on *Arabidopsis* against aphids. Based on previous reports of enhanced *Arabidopsis* resistance against pathogens due to OGs [20,23,24], we used an OG mixture of DP10–15 in this study. We showed that treatment with OGs decreased the offspring number and feeding performance of *Myzus persicae* aphids, and this was associated with an accumulation of callose, ROS, and the activation of the SA signaling pathway.

## 2. Results

### 2.1. Treatment with OGs Decreases the Offspring Number of M. persicae

The primary goal of any pesticide is to eliminate or reduce the pest population in the crop. Thus, we first assessed the effect of OGs on the aphid population. The total offspring number significantly reduced by 38% when *M. persicae* aphids were allowed to obligately feed on OGs DP10–15-treated plants (no-choice assay) than on mock-treated plants (Figure 1A). In addition, we considered that OGs DP10–15-treated plants might be less preferred as hosts by aphids. To test this hypothesis, we performed a “free-choice” assay, wherein aphids could freely settle on their preferred host based on visual, olfactory, and gustatory evaluation. We found that both groups of plants were equally preferred by *M. persicae* (Figure 1B).

### 2.2. Aphids Experience Mechanical Difficulties in Probing and Extended Times of Phloem Salivation when Feeding on OG-Treated Plants

To determine the reasons underlying the reduced aphid population in OG (DP 10–15)-treated plants (Figure 1A), we analyzed their feeding behavior at 6, 12, 24, and 72 h post-treatment (hpt) using electrical penetration graph (EPG) assays. The feeding behavior of the aphids in the presence of the two groups of plants did not significantly change at 6 and 12 hpt (Appendix A). However, at 24 hpt, aphids showed a significant 3-fold increase in the total time they underwent stylet derailment (i.e., mechanical difficulties to penetrate) in OG-treated plants compared to mock (Appendix A). In addition, at 72 hpt, this effect strengthened to a 14-fold increase when feeding on OG-treated plants with respect to the control group (Figure 2).

Moreover, at 72 hpt, we found that the time spent by aphids injecting saliva into the phloem and the time they took for the initial probe significantly increased when they fed on OG-treated than on mock-treated plants (Figure 2).

### 2.3. OGs Induce an Accumulation of Callose and ROS

Callose synthesis is a common plant defense mechanism against aphids [25,26]. Indeed, we consistently found callose deposition surrounding the stylets of *M. persicae* that infested *Arabidopsis* (Figure 3C and Appendix A). Considering that OGs elicit the deposition of callose [27], we hypothesized that this could contribute to the decreased feeding performance of aphids on OG-treated plants (Figure 1 and Figure 2). Thus, we evaluated the kinetics of callose deposition after OG treatment at the same temporalities at which EPGs were performed. OGs with DP 10–15 more efficiently elicited callose accumulation than those with DP 3 and DP 25–50 (Appendix A). They also significantly induced substantial amounts of callose deposition in *Arabidopsis* leaves compared with that observed in mock-treated plants at all timepoints (Figure 3A,B, Appendix A). Callose accumulation was higher at the later timepoints (24 and 72 hpt) than at the earlier ones (6 and 12 hpt; Figure 3A,B).

The accumulation of ROS upon aphid infestation in plants has been previously reported [28,29], and in *Arabidopsis*, it can be elicited by OG treatment, where it peaks at early temporalities (6–8 hpt) [20,30]. In our study, intracellular ROS accumulated locally, close to the feeding site of *M. persicae* both in the vascular bundle and mesophyll, 6 h post-infestation (Figure 4A and Appendix A). The intracellular ROS at 6 hpt, induced by OG treatment, was not detected in mock-treated leaves (Figure 4B,C and Appendix A).

### 2.4. OGs Increase the Expression of Genes Related to the SA Signaling Pathway

The SA signaling pathway is a critical element of the plant defense response against biotic threats. Thus, to determine if it was involved in the OG-induced resistance to aphids, we evaluated the expression profiles of the SA marker gene*PR1* and of the genes participating in hormone biosynthesis and signaling, namely, *SARD1*, *EDS1*, and *PAD4*. OG (DP 10–15)-treated plants exhibited significantly increased transcript accumulation of the four evaluated SA marker genes compared with mock-treated plants at 6 and 12 hpt (Figure 5A). *SARD1* transcripts significantly increased by 4.0-fold at 6 hpt, lasting until 24 hpt (Figure 5A). *EDS1* and *PAD4* transcripts significantly increased in number by 3.4- and 5.6-fold, respectively, at 6 hpt. *PR1* showed the most drastic and significant increase of 57-fold in transcript levels at 12 hpt (Figure 5A). The PR1 protein also evidently accumulated at 12 and 24 hpt (Figure 5B).

### 2.5. OG-Mediated Resistance Requires SA Accumulation

Concomitant with the significant increase in SA signaling and SA biosynthesis gene expression, we found that the total hormone levels also increased by 2.4-fold and 3.2-fold at 12 and 24 hpt, respectively (Figure 6A), supporting the PR1 protein accumulation in OG-infiltrated plants at the same temporalities (Figure 5B). This result further confirms that OGs (DP 10–15) activate SA signaling.

To evaluate the biological importance of SA accumulation to the OG-mediated defense against aphids, we used *Arabidopsis* plants deficient in SA accumulation, NahG plants, expressing a salicylate hydroxylase enzyme, which converts SA to catechol [31], and a *sid2* mutant plant deficient in SA biosynthesis [32]. These genotypes were first treated with OGs (DP 10–15) and then challenged with aphids in a no-choice assay to evaluate the number of offspring produced by the insect. We found that treating these plants with the OGs did not increase their resistance against *M. persicae* aphids, as no changes were observed in the total number of offspring (Figure 6B).

## 3. Discussion

The most common strategy to control aphid infestation is the use of pesticides. However, these agrochemicals pose risks to human health and the environment. Exposure to pesticides via water bodies, air, fruits, and vegetables, along with occupational exposure, has been associated with different types of cancers in adults and children [33,34,35,36,37]. The European Commission, through the “European Green Deal,” aims to reduce the use and risk of chemical pesticides by 50% by 2030 [38]. Therefore, greener pesticides and new pest management strategies are necessary for achieving sustainable and eco-friendly agricultural goals.

OGs, which are plant cell wall-derived oligomers specifically obtained from the homogalacturonan domain, are promising immune-stimulating molecules to fulfill this goal. They are extracted from fruit industry waste, such as apple byproducts and citrus peel [9,10,11]. These oligomers have been shown to participate in relevant plants mechanisms such as the trade-off between plant growth and defense. For instance, transgenic *Arabidopsis* plants with the capacity to accumulate high levels of endogenous OGs were more resistant to *B. cinerea*, *P. carotovorum*, and *P. syringae* than wild-type plants. However, they also exhibited reduced growth [39], suggesting that the constant activation of defense responses caused by the abnormally high levels of OGs consumes and diverts the resources allocated for growth. Indeed, the application of OGs downregulates the expression of auxin-related genes [40]. Thus, the strategy for using OGs as a pest control agent must be carefully developed to avoid biomass or yield reduction.

OGs have been shown to reduce disease symptoms caused by common crop pathogens, such as *B. cinerea*, in grapevines, strawberries, *Arabidopsis*, and tomatoes [12,13,41,42]. However, their protective effects against phytophagous insects such as aphids are unknown. Our study indicates that OGs (DP 10–15) effectively reduced the colonization of *M. persicae* on *Arabidopsis* leaves because they deposited significantly less offspring and encountered more difficulties feeding on OG-treated plants than on mock-treated plants (Figure 1A and Figure 2). Despite the known involvement of OGs in the defense–growth trade-off, no differences were found in the size of OG-treated plants compared with mock-treated ones (Appendix A), probably because all experiments involved treatment of mature, fully expanded leaves. Thus, the possibility that OGs affected growth responses in these leaves was reduced, and their effect was limited to their defense-eliciting activity. Intriguingly, despite decreased aphid performance, treatment with OGs (DP 10–15) did not dissuade aphids from settling (Figure 1B), suggesting that aphids cannot discriminate between OG-treated and mock-treated plants in the first stages of host finding and settling. This exciting feature of OGs could be further studied and developed to improve the efficiency of trap cropping in integrated pest management strategies. The treated trap plants could silently reduce the aphid population without compromising their attractiveness to the insects.

Further, we identified the plant defense responses triggered by OGs that could be responsible for the reduced performance of aphids. We found that callose deposition substantially accumulates after OGs DP10–15 treatments, peaking at 24 and 72 hpt (Figure 3, Appendix A), which correlates with the increase in feeding difficulties experienced by aphids at these timepoints (Figure 2, Appendix A). The substantial deposition of callose could hinder the movement of the stylets through the intercellular polymer matrix, as the aphids fed on OG-treated plants. This hypothesis is supported by the fact that increased callose synthesis has been considered a trait of resistance against aphids in some varieties of wheat, barley, and pepper [25,26,43]. Moreover, the ROS accumulation triggered by OGs (Figure 4) could also contribute to the penetration difficulties in aphids, as ROS produced during OG oxidation in the apoplast has been shown to promote cell wall fortification through lignin polymerization and indole-3-acetic acid oxidation [44].

We also observed that treatment with OGs increased the time spent by aphids on phloem salivation (Figure 2). This feeding activity had been related to the injection of salivary elements of aphids (e.g., effector proteins) that point to suppress defense mechanisms of plants [45,46]. For example, the *M. persicae* salivary protein Mp55 suppresses PTI responses in *Arabidopsis* because its expression in the plant significantly reduced callose and ROS accumulation upon aphid infestation compared with wild-type plants [47]. Thus, the longer times spent in phloem salivation while feeding on OG-treated plants could be a strategy by which aphids try to suppress OG-elicited callose deposition (Figure 3), ROS accumulation (Figure 4), and activation of the SA signaling pathway (Figure 5).

Our study highlighted that SA is a critical component driving the OG-elicited defense response against aphids. Treatment with OGs (DP 10–15) activated SA signaling and accumulation (Figure 5 and Figure 6), consistent with a previous study showing that the downstream signaling after OGs bind to the pectin receptors (WAKs) depends on *PAD4* and *EDS1* [48], which are critical genes participating in the SA-mediated immune response in *Arabidopsis* [49]. In addition, the increase in ROS and SA levels (Figure 4 and Figure 6A) is consistent with the described cross-relationship between these molecules, wherein SA accumulation is preceded by a ROS burst that activates a PTI or an effector-triggered immune response [50,51], conferring increased resistance against bacterial pathogens and, as we have demonstrated in this report, against aphids. Our results are also consistent with those of a previous study wherein treatment with OGs led to the increased transcription of SA-related genes in *Arabidopsis*, resulting in enhanced resistance against *P. syringae* [15].

To further understand the role of SA in OG-induced resistance against aphids, we treated two SA-deficient *Arabidopsis* genotypes (NahG and *sid2*) [31,32] with OGs and challenged them with *M. persicae* aphids. The results revealed that the resistance conferred by OGs in wild-type plants was abolished in NahG and *sid2* plants as treating them with OGs did not decrease the number of aphid offspring (Figure 6, highlighting that SA is essential for OG-induced resistance against aphids. Altogether, this study shows that treatment with OGs with a DP of 10–15 enhances the resistance of *Arabidopsis* against aphids (Figure 1 and Figure 2) by inducing PTI responses (callose and ROS accumulation; Figure 3 and Figure 4) and activating the SA signaling pathway (Figure 5 and Figure 6).

## 4. Materials and Methods

### 4.1. Maintenance of Plants and Insects

All experiments were conducted under the following controlled environmental conditions: 21–26 °C, 12 h/12 h dark/light cycle, 35–50% relative humidity, and light intensity of 120 μmol m^−2^ s^−1^. *Arabidopsis* seeds were sown in pots filled with a 2:1 substrate mixture of peat–vermiculite soil. For all assays, 4-week-old plants were used. Parthenogenetic colonies of the green peach aphid *M.*
*persicae* were maintained on *Brassica oleracea* plants grown in peat–vermiculite soil. Insect growth chambers were set to conditions of 25 °C, 12 h/12 h dark/light cycle, and 35–50% relative humidity. *B. oleracea* plants were changed regularly to encourage aphid growth.

### 4.2. Genotypes

The *Arabidopsis* genotypes used in this study correspond to wild-type Col-0, an NahG transgenic line expressing a salicylate hydroxylase from *Pseudomonas putida* that metabolizes SA to catechol [31], and a *sid2* mutant carrying a mutation in the isochorismate synthase 1 gene (*ICS1*), which is involved in SA biosynthesis [32].

### 4.3. Bioassays

For all bioassays, fully expanded leaves of 4-week-old *Arabidopsis* plants (wild-type Col-0) were infiltrated using a tipless syringe with 200 µg mL^−^^1^ of OGs, with a DP ranging from 10–15 (Elicityl OligoTech, Crolles, France), dissolved in ultrapure water. As the control (mock), ultrapure water was used. The infiltration method was used to ensure that the oligomers entered homogeneously into the leaves’ apoplast. These molecules have a large negative charge, and if sprayed, can make diffusion and entry into plant tissue difficult [52].

No-choice assays were used to evaluate the total number of offspring produced by aphids under the different conditions studied. For this, two 1-day-old nymphs of *M. persicae* aphids were placed on the treated leaves using a paintbrush and confined in transparent plastic cages with a transparent mesh on top to allow for gas exchange (Figure 1A). The total number of offspring was recorded 12 days after confinement. The results are expressed as the total number of offspring.

Free-choice assays were performed to evaluate the settling preference of aphids. First, plant-attached leaves, OG-treated or mock-treated, were placed on the choice arena as indicated in Figure 1B. Thirty wingless *M. persicae* adults were placed in the center of the choice arena and were allowed to freely choose their preferred host. After 6, 12, 24, and 72 hpt, the number of aphids on each leaf was registered. Host preferences of aphids were expressed as percentages.

Probing behavior on host plants was monitored using EPG. Briefly, aphids and plants were made parts of an electrical circuit, which is completed when the aphid inserts its stylet into the plant. Adult wingless *M. persicae* aphids were immobilized on a pipette tip coupled to a vacuum pump; then, a 12 mm diameter gold wire was attached to the insect dorsum using water-based silver glue (EPG Systems, Wageningen, the Netherlands). The other end of the gold wire was attached to the EPG probe provided with the Giga-8 device. The EPG circuit was completed by inserting a copper electrode into the plant soil (ground). Wired aphids were placed on treated leaves, and the feeding behavior was monitored for 8 h. The number of replications for each condition (mock or OGs) was at least 15. EPG waveforms were recorded using a Giga-8 DC-EPG device (EPG Systems, Wageningen, the Netherlands) and manually analyzed using Stylet1 (EPG System, Wageningen, the Netherlands. The parameters derived from EPGs were expressed as the total duration in minutes.

### 4.4. Callose and ROS Visualization

For callose imaging, the collected leaves were fixed in a solution of acetic acid and ethanol (1:3) for 1 week. They were washed twice with a solution of 0.15 M dipotassium hydrogen phosphate (K_2_HPO_4_) for 15 min and stained with an aniline blue solution for 2 h, prepared by dissolving 1 µg mL^−1^ of aniline blue dye in 100 mL of 0.15 M K_2_HPO_4_. The stained leaves were mounted on glass slides with glycerol and visualized using confocal microscopy (TCS LSI confocal microscope, Leica). The fluorescence signal intensity was used as a semi-quantitative parameter to plot the differences in callose deposition between OG- and mock-treated plants using the ImageJ software.

Intracellular ROS were visualized using the dichlorofluorescein diacetate (DCF-DA) method. The collected leaves were submerged in a solution of 0.4 mM DCFH-DA (Sigma-Aldrich, Saint Louis, MO, USA) in phosphate buffer (20 mM, pH 7.2) and 0.02 % Tween 20 (Biotium, Fremont, CA, USA) for 15 min, briefly washed with phosphate buffer (20 mM, pH 7.2), and mounted on glass slides with glycerol. Samples were visualized using confocal microscopy (TCS LSI confocal microscope, Leica Microsystems, Wetzlar, Germany).

### 4.5. Gene Expression Analysis Using RT-qPCR

Total RNA was extracted from frozen mock-treated or OG-treated *Arabidopsis* leaves using TRIzol^®^ (Invitrogen) according to the manufacturer’s instructions. The quality of RNA was evaluated using denaturing gel electrophoresis, and the RNA was quantified by measuring the absorbance at 260 and 280 nm in a spectrophotometer (EPOCH). cDNA was synthesized from 1 µg of total RNA using the All-In-One 5X RT MasterMix kit (abm, Vancouver, Canada). Then, qPCR was performed using the Brilliant III Ultra-Fast SYBR^®^ Green QPCR Master mix reagent (Agilent, Santa Clara, CA, USA, #600882) on an AriaMx real-time PCR system. The expression levels of target genes were calculated relative to the yellow leaf specific 8 gene (*YLS8*) as a reference.

### 4.6. Western Blot Analysis

To evaluate PR1 protein levels in OG-treated *Arabidopsis* leaves, total proteins were extracted, 45 µg were loaded onto 12% denaturing polyacrylamide gel, electrophoresed, and transferred to polyvinylidene difluoride membranes. Membrane blocking was performed at room temperature for 1 h using a phosphate-buffered saline–Tween 20 solution with 5% skim milk. The PR1 protein was detected using a polyclonal anti-PR1 antibody (1:1000 dilution, Agrisera, Umea, Sweden #AS10 687). ACTIN was detected using an anti-ACT primary antibody (1:5000 dilution, Agrisera, Umea, Sewden, #AS13 2640). Anti-rabbit (Thermofisher, Waltham, MA, USA, #31460) antibody was used as the secondary antibody (1:10,000 dilution). The membrane was visualized using the Pierce^®^ ECL Western Blotting Substrate (Thermofisher, Waltham, MA, USA, #32109) according to the manufacturer’s instructions.

### 4.7. SA Extraction and Quantitation Using High-Performance Liquid Chromatography (HPLC)

SA extracted from leaf tissues was quantified through HPLC using a C8 column and a fluorescence detector, as described previously [53]. Each sample consisted of 0.3–0.5 g of fresh tissue collected from a pool of mature leaves from six plants grown in soil treated as described above. Untreated plants were used as control.

### 4.8. Statistical Analysis

For all assays, a completely randomized design was used to assign individual plants to the experimental treatments. Data distributions were tested using the Shapiro–Wilk test and Levene’s test for homogeneity of variance. Normally distributed data were analyzed using the Student’s *t*-test, whereas non-normally distributed data were analyzed using the non-parametric Mann–Whitney U test. Prior to analysis, percentage data obtained from free-choice assays were transformed using ArcSin (√x). Signal intensity data from callose imaging were analyzed using two-way analysis of variance using the general linear model procedure to determine the effects of treatment (OGs), time (hours post-treatment), and their interaction. Mean separation was calculated using Fisher′s least significant difference test. EPG parameters describing probing behavior were calculated manually and the mean and standard error were subsequently calculated using Excel. The relative gene expression presented in the results corresponds to the ΔCT value, calculated as the difference between the expression of the target gene and reference gene (*YLS8*). For each gene, the relative expressions determined in the mock- and OG-treated samples were compared using the Student’s *t*-test. Statistical data were analyzed using R version 4.2.0 (R Core Team, 2022) and GraphPad Prism 8 software.

## Figures and Tables

**Figure 1 ijms-23-09753-f001:**
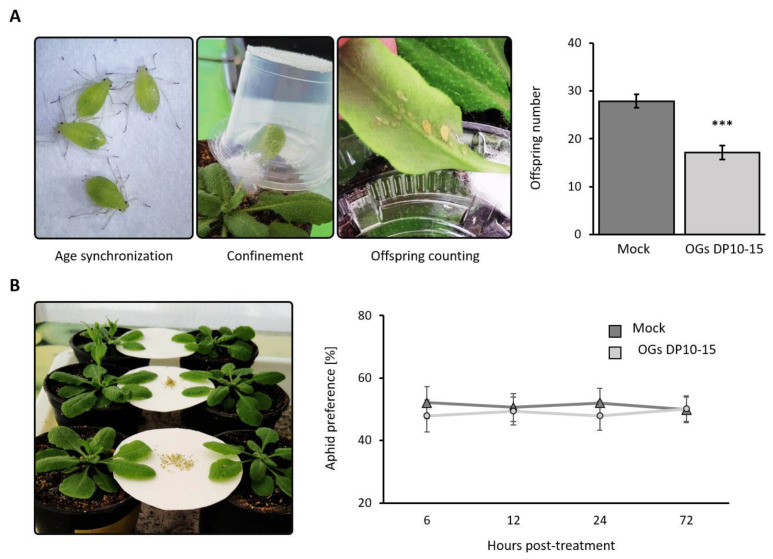
Treatment with OGs decreases the number of aphid offspring. (**A**) A no-choice assay was performed to determine the effect of OG treatment on the number of offspring deposited by *Myzus*
*persicae* aphids. Fully expanded leaves of 4-week-old *Arabidopsis* plants were infiltrated with OGs having a degree of polymerization of 10–15 (DP 10–15) (200 µg mL^−1^) or ultrapure water (mock). One-day-old nymphs were confined to the treated leaves, and 12 days after treatment, the offspring were counted. Error bars correspond to standard error. Asterisks represent significant differences determined using Student’s *t*-test (***, *p* < 0.001; *n* = 6). (**B**) A free-choice assay was executed to determine the settling preference of *M. persicae.* Fully expanded leaves of 4-week-old *Arabidopsis* plants were infiltrated with OGs (DP 10–15; 200 µg mL^−1^) or ultrapure water (mock). Subsequently, 30 adult wingless aphids were placed between the treated leaves to freely choose their preferred host at different post-treatment temporalities. The Student’s *t*-test was used for comparing means. The error bars correspond to standard error. No significant differences were found at any of the temporalities assayed (*p* > 0.05; *n* = 10).

**Figure 2 ijms-23-09753-f002:**
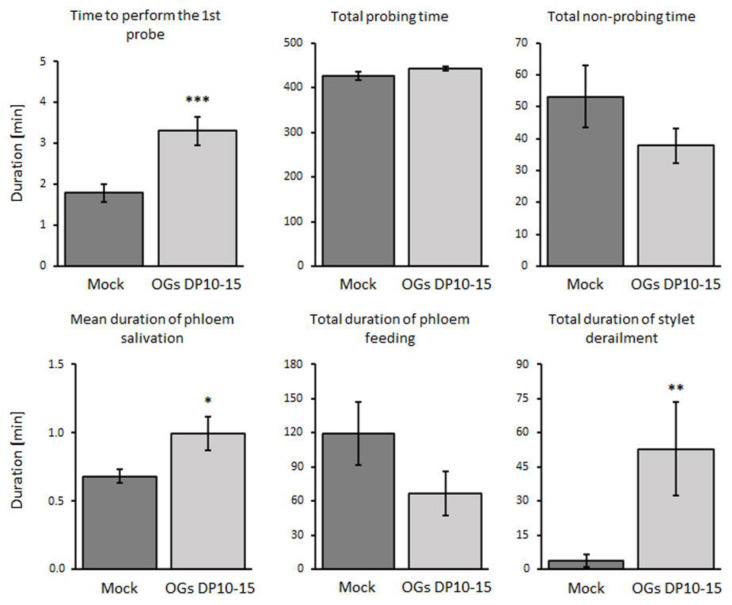
Treatment with OGs negatively alters the feeding performance of aphids. The feeding profiles of *M. persicae* aphids, on either OG- or mock-treated plants, were analyzed using electrical penetration graph (EPG) assays. Fully expanded leaves of 4-week-old *Arabidopsis* plants were infiltrated with OGs (DP 10–15; 200 µg mL^−1^) or ultrapure water (mock). One adult wingless aphid was placed over the treated leaves, and its feeding profile was recorded using EPG for 8 h. Only data from the 72 hpt timepoint are shown, as the main differences were found at this temporality. The results obtained at 6, 12, and 24 hpt are presented in Appendix A. Significant differences, indicated by asterisks, were determined using the Student’s *t*-test or the Mann–Whitney U test, depending on the data distribution of each feeding parameter (i.e., Gaussian or non-Gaussian, determined using the Shapiro–Wilk test). The error bars correspond to standard error. At least 15 independent replicates were assayed for each condition (*, *p* < 0.05; **, *p* < 0.005; ***, *p* < 0.001).

**Figure 3 ijms-23-09753-f003:**
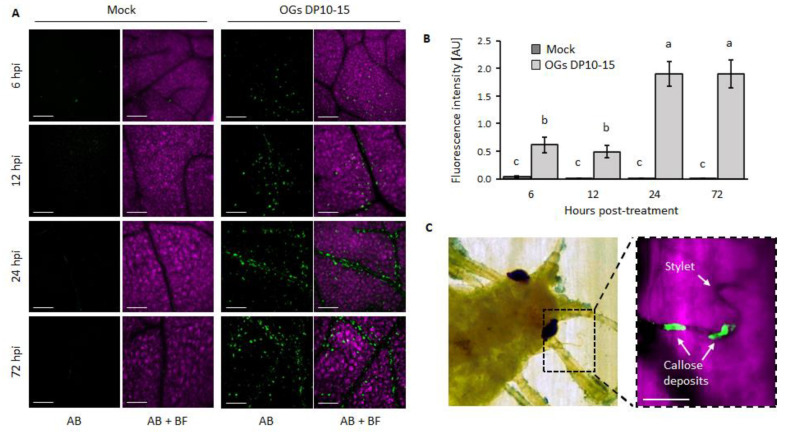
OGs induce the deposition of callose. (**A**) Representative images of callose deposits in mock- and OG-treated *Arabidopsis* plants. Fully expanded leaves of 4-week-old *Arabidopsis* plants were infiltrated with OGs (DP 10–15; 200 µg mL^−1^) or ultrapure water (mock) (*n* = 8); scalebar = 200 µm. More replicates are shown in Appendix A. (**B**) Plots show the fluorescence intensities measured from the figures in panel A using the ImageJ software. The fluorescent signal corresponds to callose deposits detected via aniline blue (AB) staining. The error bars correspond to standard error. Means followed by the same letter were not significantly different at *p* ≤ 0.05 by Fisher’s least significant difference test. (**C**) Callose deposits consistently formed at penetration sites surrounding the stylets of *M. persicae* after 6 h of feeding on an *Arabidopsis* leaf; scalebar = 50 µm. AB = aniline blue signal, BF = brightfield. More replicates are shown in Appendix A.

**Figure 4 ijms-23-09753-f004:**
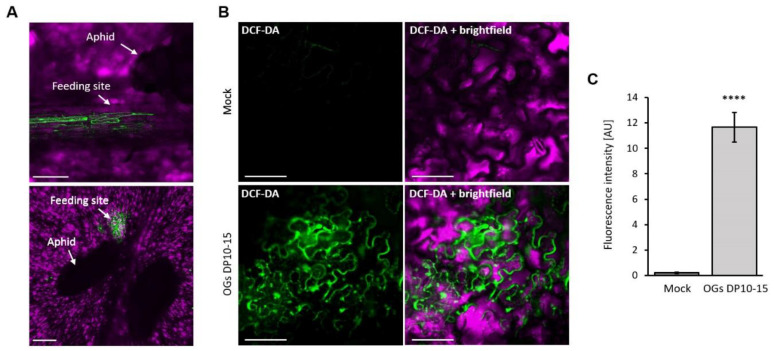
OGs induce the accumulation of intracellular reactive oxygen species (ROS). Intracellular ROS were detected using the dichlorofluorescein diacetate (DCF-DA) method. (**A**) Intracellular ROS accumulated after *M. persicae* infested *Arabidopsis* leaves. The upper image shows ROS in cells of the main vascular bundle at the feeding site of an aphid. The lower image shows ROS in mesophyll cells surrounding the feeding site of an aphid; scalebar = 300 µm. More replicates are shown in Appendix A. (**B**) Representative images of intracellular ROS accumulation in mock- and OG-treated *Arabidopsis* leaves at 6 hpt (*n* = 10); scalebar = 50 µm. More replicates are shown in Appendix A. (**C**) The plot illustrates the fluorescence intensities measured from figures in panel B calculated using the ImageJ software. The fluorescent signal corresponds to intercellular ROS detected using the DCF-DA method in mock- and OG-treated plants. Asterisks represent significant differences determined using the Student’s *t*-test (****, *p* < 0.0001).

**Figure 5 ijms-23-09753-f005:**
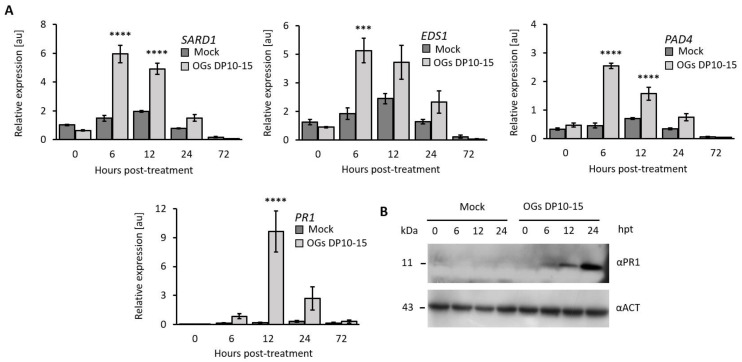
Treatment with OGs increases the expression of salicylic acid (SA)-related genes. (**A**) The transcriptional profile of SA-related genes in fully expanded leaves of 4-week-old *Arabidopsis* plants, treated with OGs (DP 10–15; 200 µg mL^−1^), at different post-treatment temporalities. Results are expressed as the relative expression in arbitrary units (au). The error bars represent standard error. Asterisks represent significant differences between mock- and OG-treated samples at the same temporalities as determined using the Student’s *t*-test (***, *p* < 0.001; ****, *p* < 0.0001; *n* = 4). (**B**) PR1 protein levels in mock- and OG-treated *Arabidopsis* leaves at different post-treatment temporalities analyzed using Western blot.

**Figure 6 ijms-23-09753-f006:**
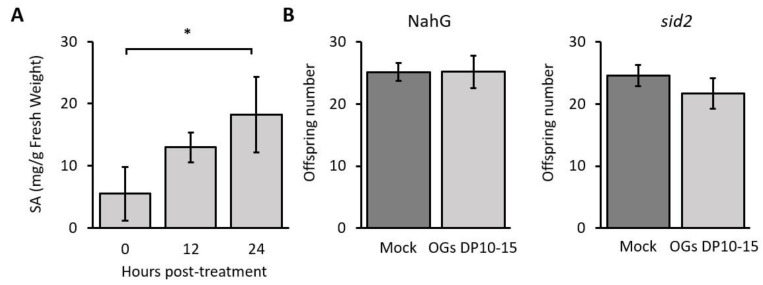
OG-induced resistance against aphids requires SA accumulation. (**A**) SA content measured in *Arabidopsis* leaves treated with OGs (DP 10–15) at 0, 12, and 24 hpt (*n* = 3). Asterisk represents significant differences found using Student’s *t*-tests (*, *p* < 0.05). (**B**) No-choice assay performed with NahG and *sid2 Arabidopsis* genotypes deficient in SA accumulation. Fully expanded leaves of 4-week-old *Arabidopsis* plants were infiltrated with OGs (DP 10–15; 200 µg mL^−1^) or ultrapure water (mock). One-day-old nymphs were confined to treated leaves of both NahG and *sid2* plants, and 12 days after treatment, the number of offspring was counted. Using Student’s *t*-test, no significant differences were found between the number of offspring deposited by aphids on mock- and OG-treated NahG and *sid2* plants. At least six replicates were assayed for each condition.

## Data Availability

The data presented in this study are available in Appendix A.

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
