# Peer review of "Oligogalacturonides Enhance Resistance against Aphids through Pattern-Triggered Immunity and Activation of Salicylic Acid Signaling"

_ijms, 2022, doi:10.3390/ijms23179753_

Round 1

Reviewer 1 Report

The manuscript entitled "Oligogalacturonides enhance aphid resistance through Pattern-Triggered Immunity and activation of salicylic acid signaling" by Silva-Sanzana et al. described the function of Oligogalacturonides (OGs) on mitigating aphis damage in plants. The study also demonstrated the increase resistance to aphis infestation was attributed to reduced number of insect offspring and changed feeding performance. The authors further revealed that OGs increased the expression of genes related to the SA signaling pathway, facilitated accumulation of callose deposits and ROS.  Overall, the paper is well written. I suggest a minor revision before it's publication. 

Comments:

1. Please discuss if OGs application affect plant growth per se.

2. Please explain why using infiltration but not other methods for applying OGs to plants, like spray onto foliage or as irrigation. 

3. Please check throughout the manuscript and use Italics for species latin names.

Reviewer 2 Report

Manuscript ID: ijms-1866761 

Manuscript title: Oligogalacturonides enhance aphid resistance through Pattern-Triggered Immunity and activation of salicylic acid signaling

Authors: Christian Silva-Sanzana, Diego Zavala, Felipe Moraga, Ariel Herrera-Vásquez, Francisc Blanco-Herrera

The present manuscript deals with the protective response of exogenous oligogalacturonides towards the infestation brought about by the green peach aphid, Myzus persicae.on Arabidopsis.

The manuscript is interesting, rather well written and easy to follow. The methodology seems to be appropriate. The results are presented in a clear manner and appropriately discussed, also on the background of the available literature.

There are, in my opinion, a couple of major points, and some minor ones, all of which need to be appropriately and adequately addressed on the part of the Authors.

    1)    Statistics: although adequate description is provided concerning biological and technical replication for some of the parameters measured, there are cases in which more details are needed. For example, how many individual plants have been used for the feeding assays? Has a statistical design been followed, to assign the individual plants to the experimental treatments?

    2)  Results and Discussion: since the Results section of the present manuscript does not contain the mere reporting of the results obtained, as a conventional Results section should do, but indeed it contains also elements more appropriate to a discussion, it would  be better, in my opinion, unifying the two sections into a unique Results & Discussion section (requiring in such a case a short Conclusions section at the end). This would also avoid a certain degree of repetition of the same concepts, at times.

3  3) What about ROS?: although ROS have been measured here, and their results reported in the figures, no comment is made here concerning their role in plant-host interaction and about their relationships with the other measured parameters (e.g. SA). Just as an example: what is the functional meaning, in the context of plant-host interaction, of an increase in ROS level following an exogenous OG supply?

Minor points:

4  4) Please use the exponential notation throughout, e.g. mg mL-1, instead of the fractional one, e.g. mg/mL

5   5) Please check all the acronyms and ensure that the uncommon ones are spoken out in full, upon first mentio

6  6) Please check italics, to be used throughout for Linnean names, e.g. M. persicae, and for genes, too, including mutants

All the above considering, I recommend major revision of the present manuscript, accurately addressing all the points raised above.

Round 2

Reviewer 2 Report

In the present revised version the Authors have adequately addressed all the points raised from my side on their original submission